# Different Patterns of HIV-1 Replication in MACROPHAGES is Led by Co-Receptor Usage

**DOI:** 10.3390/medicina55060297

**Published:** 2019-06-21

**Authors:** Ana Borrajo, Alessandro Ranazzi, Michela Pollicita, Maria Concetta Bellocchi, Romina Salpini, Maria Vittoria Mauro, Francesca Ceccherini-Silberstein, Carlo Federico Perno, Valentina Svicher, Stefano Aquaro

**Affiliations:** 1Department of Experimental Medicine and Surgery, University of Rome Tor Vergata, 00133 Roma, Italy; ana.borrajo@hotmail.com (A.B.); aranazzi@yahoo.com (A.R.); michela.pollicita@uniroma2.it (M.P.); mariac.bellocchi@gmail.com (M.C.B.); rsalpini@yahoo.it (R.S); ceccherini@med.uniroma2.it (F.C.-S.); 2Group of Virology and Pathogenesis, Galicia Sur Health Research Institute (IIS Galicia Sur)-Complexo Hospitalario Universitario de Vigo, SERGAS-UVigo, 36312 Vigo, Spain; 3Department of Microbiology and Virology, Complex Operative Unit (UOC), Hospital of Cosenza, 87100 Cosenza, Italy; m.v.mauro@virgilio.it; 4Department of Microbiology and Clinic Microbiology, University of Milan, 20162 Milan, Italy; cf.perno@uniroma2.it; 5Department of Pharmacy, Health and Nutritional Sciences, University of Calabria, 87036 Rende, Italy

**Keywords:** α chemokine receptor 4, β-chemokine receptor 5, human immunodeficiency virus, monocyte-derived macrophages

## Abstract

*Background and objectives:* To enter the target cell, HIV-1 binds not only CD4 but also a co-receptor β-chemokine receptor 5 (CCR5) or α chemokine receptor 4 (CXCR4). Limited information is available on the impact of co-receptor usage on HIV-1 replication in monocyte-derived macrophages (MDM) and on the homeostasis of this important cellular reservoir. *Materials and Methods:* Replication (measured by p24 production) of the CCR5-tropic 81A strain increased up to 10 days post-infection and then reached a plateau. Conversely, the replication of the CXCR4-tropic NL4.3 strain (after an initial increase up to day 7) underwent a drastic decrease becoming almost undetectable after 10 days post-infection. The ability of CCR5-tropic and CXCR4-tropic strains to induce cell death in MDM was then evaluated. While for CCR5-tropic 81A the rate of apoptosis in MDM was comparable to uninfected MDM, the infection of CXCR4-tropic NL4.3 in MDM was associated with a rate of 14.3% of apoptotic cells at day 6 reaching a peak of 43.5% at day 10 post-infection. *Results:* This suggests that the decrease in CXCR4-tropic strain replication in MDM can be due to their ability to induce cell death in MDM. The increase in apoptosis was paralleled with a 2-fold increase in the phosphorylated form of p38 compared to WT. Furthermore, microarray analysis showed modulation of proapoptotic and cancer-related genes induced by CXCR4-tropic strains starting from 24 h after infection, whereas CCR5 viruses modulated the expression of genes not correlated with apoptotic-pathways. *Conclusions:* In conclusion, CXCR4-tropic strains can induce a remarkable depletion of MDM. Conversely, MDM can represent an important cellular reservoir for CCR5-tropic strains supporting the role of CCR5-usage in HIV-1 pathogenesis and as a pharmacological target to contribute to an HIV-1 cure.

## 1. Introduction

Combined antiretroviral therapy (cART) does not eradicate HIV-1 [1,2] due to the early establishment of a long-lived viral reservoir [3,4,5,6]. This reservoir can include cells of macrophage lineage where, in contrast to CD4+ lymphocytes, HIV is relatively non cytopathic and can replicate extensively in intracellular compartments in a long-lasting manner [7,8,9,10]. HIV-1 infected monocyte-derived macrophages (MDM) are fully capable of producing infectious viral particles when cART is discontinued [11,12,13,14,15,16] and may play a key role in regulating the disease progression [17].

Over the following decades, after the discovery of CD4 as the main virus receptor [18,19], further studies have demonstrated that the chemokines coreceptors CCR5 and CXCR4 play crucial duties in supporting infection of HIV-1 in target cells.

Binding of chemokine receptors CCR5 or CXCR4 is widely thought to be the cause that stimulates the membrane fusion during HIV-replicative life cycle [19,20]. Infection with HIV-1 is generally initiated by macrophages, slowly replicating, non-syncytium-inducing (NSI) variants [20,21] that utilize CCR5 as a coreceptor [22,23,24]. In 50% of instances, disease evolution is correlated with the development of syncytium-inducing (SI) variants which at least use CXCR4 [25,26,27,28].

The tropism of HIV-1 for specific and relevant cell populations in diverse compartments is determined by the coreceptor utilized by HIV-1 Env for the entrance of the viral particles [28]. For infection of MDM cultures, HIV viruses preferentially utilize CCR5 as a coreceptor [22,23,24,25,26], whereas viruses in T-cells use CXCR4 [27]. Dual-tropic viruses can utilize both coreceptors (CCR5/CXCR4) [29,30]. Thus, the coreceptor particularity of primary HIV-1 isolates is commonly utilized to characterize cellular tropism [31].

Previous studies have shown that CCR5 is present on a wide variety of cells that can be infected by HIV-viruses, including T cells, monocytes and macrophages. A lot of research has shown the existence of CCR5-tropic viruses, which were proficient in replication of primary CD4+ T cells but which could not effectively infect MDM [32,33,34,35,36,37,38]. Also, some CCR5-tropic primary HIV-1 strains utilize CXCR4 for input into MDM [32,38]. Hence, the viral determinants that regulate HIV-1 tropism for macrophages are considerably more complicated than the coreceptor specificity of the virus. HIV-1 viruses use CCR5 for their infection, although their primary targets are T cells not macrophages. It is widely agreed that these CCR5 and CXCR4 viruses can replicate in both macrophages as well as in T cells. However, their replication effectiveness changes in different cell classes which depend upon the cellular environment [37,38]. Moreover, viral progeny from macrophages and T cells may have divergent groups of host protein integrated in their viral particle [39].

HIV-mediated patterns of replication in latently infected cells (virus reservoir) have not been completely understood. HIV infections lead to increased expression of specific proteins like B-cell lymphoma 2 (BCL-2), B-cell lymphoma-extra large (BCL-XL), cellular FLICE (FADD-like IL-1β-converting enzyme)-inhibitory protein (cFLIP), Induced myeloid leukemia cell differentiation protein (Mcl-1) [40,41,42] or downregulation of proteins Bcl-2-associated X protein (BAX), Bcl-2-associated death promoter protein (BAD proteins), Fas-associated protein with death domain (FADD) [42,43,44]. These factors contribute to regulate the transcription of genes correlated with host defense, cellular anti-oxidant molecules like glutathione and thioredoxin, signal transduction, survival, and the cell cycle, including the cyclin-dependent kinase inhibitor 1A (CDKN1A/p21) gene whose maximum extent of mRNA and protein expression parallels active HIV-1 replication in latent cells.

This work aims at defining: (i) the role of CCR5-tropic and CXCR4- tropic strains in MDM; (ii) assessment of different patterns of replication in this cell type by evaluating the extent of DNA degradation, viral production, p38 MAPK activation and survival gene modulation in CXCR4 and the CCR5 infected MDM.

## 2. Materials and Methods

### 2.1. Virus

HIV-1 pNL4-3p10-17 and p81Ap10-17 molecular constructs were obtained from B. Chesebro, (National Institute of Allergy and Infectious Diseases, Hamilton, Montana 59840, USA) and contain the whole HIV-1 genome [45]. The HIV-1 p81A p10-17 clone was generated by replacing the pNL4-3p10-17 a 659 bp long sequence of env. This nucleotide sequence belongs to the R5-tropic HIV-1 Ba-L and includes the V1, V2, V3 variable domains, whereas NL4-3 is a CXCR4-tropic strain, 81A replicates in cells of the MDM lineage. These plasmids were transfected in 293T through the FuGENE 6TM (Roche), a lipidic, not liposomial reagent.

HIV-1 clinical isolates #17 (X4), #6 (R5) and #10 (R5/X4) were obtained from patients enrolled from the Katholieke Universiteit Leuven (Rega institute, Leuven, Belgium) and expanded in peripheral blood mononuclear cells (PBMC).

The laboratory-adapted HIV-1 X4 strain IIIB was expanded in H9 cells and obtained from supernatants at day 8 post infection. The laboratory-adapted HIV-1 adapted R5-tropic strain Ba-L was expanded in MDM [46,47].

All the strains were purified from supernatants of the respective cultures after centrifugation at 20,000 rpm for 2 h, filtered through 0.45 µm filter, DNase I treated, and concentrated with a Centricon Plus-20 membrane with a 100,000 molecular weight cut-off (Millipore Corporation, Bedford, Mass.) to remove contaminating cytokines and growth factors which might interfere with signal transduction analysis. Concentrated virus was stored in aliquots at −70 °C until use. Stock virus titers were determined with a colorimetric reverse transcriptase activity assay (Roche Molecular Biochemicals, Indianapolis, USA).

### 2.2. Drugs

The prototype bicyclam CXCR4 inhibitor and agonist stromal derived factor (SDF-1alpha) AMD3100, (1-1*-[1,4-phenylenebis(methylene)]-bis(1,4,8,11-tetraazacyclotetradecane) octahydrochloride dihydrate]) synthesized at Johnson Matthey [48,49,50], and the CCR5 inhibitor, N,N-dimethyl-N-[4-[[[2-(4-methylphenyl)-6,7-dihydro-5H-benzocyclohepten-8-yl] carbonil]amino]benzyl] tetrahydro-2H-pyran-4-aminium chloride (TAK779), a nonpeptide compound with a small molecular weight (Mr 531.13), (Takeda Chemical Industries, Ltd., Osaka, Japan) [51,52], were suspended and aliquoted in Phosphate-buffered saline (PBS) solution, and used to 5 µM and 2 and 10 µg/mL, respectively.

### 2.3. Cells

Human primary MDM were generated and purified as previously described [53,54,55,56]. MDM were derived from PBMCs of healthy donors. Briefly, PBMCs were separated by Ficoll-Hypaque gradient centrifugation and seeded in T25 flasks at a number of 50.10^6^ cells in 7 mL Roswell Park Memorial Institute (RPMI) medium 1640 supplemented with 20% heat inactivated, mycoplasma- and endotoxin-free fetal bovine serum (FBS), L-glutamine (1 mM), penicillin (100 U/mL), and streptomycin (100 µg/mL), without exogenous cytokines or growth factors, at 37 °C in a humidified atmosphere enriched with 5% CO2. After five days of culture, non-adherent cells were eliminated with caution by consecutive gentle washings with warmed RPMI 1640, leaving a monolayer of adherent cells which were finally incubated in complete medium [57,58,59].

The MDM obtained showed a purity exceeding 98% as tested by cytofluorimetric analysis. Expression of CXCR4 and CCR5 in all our MDM cultures was assayed by flow cytometric analysis (FCM) (FACScanTM, Becton Dickinson System, San José, CA) by means of CD184 (CXCR4/fusin) R-phycoerythrin (R-PE)-conjugated mouse anti-human monoclonal antibody and CD195 (CCR5) R-phycoerythrin (R-PE)-conjugated mouse anti-human monoclonal antibody both purchased from BD Pharmingen (Becton Dickinson biosciences, USA). Measurements were performed in at least 3 independent experiments. In each experiment, MDM derived from a single healthy donor.

### 2.4. Drug Treatment, Infection and Virus Detection

Six days after Ficoll-Hypaque, non-adherent cells were removed, and monocytes were further allowed to differentiate in MDM for four days. Their purity exceeded 98%. For exposure to inhibitor AMD3100 and TAK779, at least twenty-four hours before CXCR4 and CCR5 strain infection, MDM culture medium was removed and replaced with fresh media 20% serum. 45–60 min before infection; where needed, the drug was added to the cell supernatants at appropriate concentrations (0.4–2 µg/mL for TAK779, and 5 µM for AMD3100) and then MDM were reincubated at 37 °C in a humidified atmosphere enriched with 5% CO2.

Virus challenge was performed for at least 4 h to almost a week by exposing MDM to 3000 up to 7500 pg/mL of p24 (corresponding to, respectively, 400 and 1000 tissue cultures infectious doses 50% per ml (TCID50/mL) of the Laboratory-adapted strain HIV-1 Ba-L) of the all strains described above, followed by extensive washing to remove excess virus.

Virus production was assessed by the HIV-1 p24 gag antigen concentration in culture supernatants using a p24 gag antigen detection kit according to the instructions of the manufacturer (Abbott labs, Pomezia, Italy).

### 2.5. Western Blotting of Cell Cultures

Cells were challenged with IIIB, NL4-3, Ba-L and 81A strains of HIV (whole visions) in warmed media 20% serum at indicated times at 37 °C in a humidified atmosphere enriched with 5% CO_2_, incubated for the times indicated and then lysed and subjected to immunoblot analysis. Lysis was performed in ice-cold buffer Radio-Immunoprecipitation Assay (RIPA) (50 mM tris hydroxymethyl aminomethane ((Tris)-HCl), pH 7.4; 250 mM NaCl; 50 mM NaF, EDTA 5 mM; 0,15% Triton X-100) containing a protease and phosphatase inhibitor cocktails (1 mM phenylmethylsulfonyl fluoride; 10 µg/mL pepstatin; 10 µg/mL leupeptin and 1 mm sodium vanadate) and incubated for different times at 4 °C. Cell lysates were then clarified by centrifugation at 13,000 rpm for 10 min at 4 °C.

Protein concentrations were determined by a spectrophotometric assay (Pierce). Immunoblot analysis was performed on cell lysates containing 30 µg protein mixed with Laemmli buffer and boiled for 5 min. Samples were subjected to 10% Sodium dodecyl sulfate-polyacrylamide gel electrophoresis (SDS-PAGE) and transferred to nitrocellulose membranes. The membranes were blocked overnight with 5% Bovine serum Albumine (BSA) in TBS-tween.

A 1:1000 dilution of the lysates was used for the detection of activated MAPK/p38 a polyclonal antibody specific for Phospho-p38 (Thr202/Tyr204), and for total p38 (Cell Signaling Technology, Beverly, MA). (Jackson ImmunoResearch Laboratories). Then, membranes were treated with the corresponding horseradish peroxidase (HRP)-conjugated secondary antibody (1:5000 dilution) (Jackson ImmunoResearch Laboratories). The immunoreactive bands were visualized using enhanced chemiluminescence Western blotting system (Immun-Star HRP Chemiluminescent Kit, Hercules, Ca, USA) according to the manufacturer’s instructions (Biorad, Hercules, CA, USA).

Blots were stripped (2% SDS, 62.5 mM Tris, 100 mM mercaptoethanol) for 30 min at 56 °C and washed in PBS containing 0.05% Tween 20, before blocking and reprobing with primary antibody. For the quantification of the phosphorylated and total proteins, the bands on the films were first scanned by the Epson software program and then the images were processed through the Scion Image analysis program (Houston, TX, USA) for the IBM PC based on the popular NIH Image on the Macintosh platform.

### 2.6. RNA Isolation and Microarray Analysis

In different experiments, 81A and NL4-3 infected MDM were incubated in parallel at a dose of 2000 pg/mL of viral p24 from 6 to 24 h of infection, then exposed to 4M guanidinium and total RNA was isolated by the guanidinium-phenol procedure. The isolation of polyA mRNA from each total RNA preparation was obtained by OLIGOTEX mRNA Purification System (Qiagen, Hilden, Germany). cDNA probes for microarray experiments were prepared from 0.5 μg of cellular mRNA by CyScribe post-Labelling Kit (Amersham Biosciences). The CyDye labelled cDNA was purificated by QIAquik PCR purification Kit (Qiagen). For dual colour hybridization we combined Cy3 and Cy5 labelled cDNAs in one tube. The solution protected from light was dried by using a rotary evaporator and adding tRNA (40 γ), Calf Thymus DNA (40 γ) and Cot-1 DNA (1 γ). The dry solution was dissolved in nuclease free water, denatured at 95 °C for 5 min and cooled in ice. Hybridization buffer 4X (supplied in the CyScribe Post-labelling Kit) was added with ½ volume of 100% formammide. Hybridization was performed in an humid hybridization chamber (5X SSC) at 42 °C for 14–18 h, following washing with saline-sodium citrate (SSC) and SDS (*w*/*v*) pre-warmed to 37 °C. Fluorescent-array images were collected for both Cy3 and Cy5 by using a ScanArray Express, Microarray Analysis System Version 2.0 (Perkin-Elmer) and image intensity data were extracted and analysed by using QuantArray Pachard Biochips Software. In particular, QuantArray Software provides automated analysis of color microarray images (automatic scanning and quantitation to measure fluorescence signal at each spot on the array) before exporting data to bioinformatics software packages. Triplicate array positions are used for each gene to avoid signal noise. The human Cancer Chip version 4.0 (Takara) slides were used for microarray analysis and all spots were known. In order to evaluate the inhibition or enhancement of genes expression in terms of mRNA production, a comparison of Cy3 and Cy5 signals intensity was applied.

### 2.7. Flow Cytometry Measurement of Apoptotic Cells

At established time points of infection (see results), MDM were washed and detached from the 25 flasks with gentle scraping as previously described [60]. MDM were precipitated by centrifugation at 1500 rpm for 5 min at 4 °C. All MDM in the culture, both adherent and non-adherent, were considered in the final count for apoptosis analysis by Flow Cytometry measurement (FCM). Supernatants were removed, then aliquoted, and stored at −80 °C for p24 titration. After washing, MDM were kept in 3 mL of cold PBS, 0.02% EDTA in 10 min at 4 °C, then gently scraped and transferred to the respective tubes and precipitated by centrifugation at 1500 rpm for 5 min at 4 °C. Supernatants were removed and the pellet resuspended in 0.5 mL of Trypan-blue solution for cell count and viability check. Cells were washed with 2 mL PBS cold and centrifuged as described above.

The supernatants were completely removed and 2 mL of 70% ice ethanol was added to permeabilize for 40 min at 4 °C. After washing, cells were centrifuged (1600 rpm/5 min/4 °C), again washed and centrifuged as described above and gently resuspended in 1.0 mL hysotonic Propidium Iodide (PI) solution (Sigma) 50 µg/mL, in PBS and RNase 50 µg/mL) in polipropylen tubes. After rotating for 15 min at room temperature, the tubes were placed at 4 °C for 2 h in the dark. Cells were washed with 2 mL PBS cold, centrifuged at 1500 rpm for 5 min at 4 °C, resuspended in 0.4 mL cold PBS and kept in the dark at 4 °C for not more than 20 min before PI fluorescent measurement. The DNA specific fluorocrome Propidium Iodide (PI) recognized apoptotic cells as a distinct hyploploid cell population with a reduced staining below the G0/G1 population of normal diploid cells as results of cell shrinkage, nuclear condensation, internucleosomal DNA fragmentation [61]. The PI fluorescence was measured by Flow Cytometry in FL2-H (FACScanTM, Becton Dickinson System, San Josè, CA, USA) and registered on a logarithmic scale. All the tests were performed in duplicate.

### 2.8. Statistics

Differences were considered statistically significant at *p* ≤ 0.05 by means of a Chi Square test of independence based on a 2 × 2 contingency table. Statistical analyses were carried out with SigmaStat 3.0 (Jandel Scientific, San Rafael, CA, USA).

## 3. Results

### 3.1. Different Replicative Kinetics of CXCR4- and CCR5-Dependent HIV Strains in MDM

The first step of this study was to evaluate the efficiency of CCR5-, CXR4- tropic HIV-1 strains to replicate in MDM by measuring p24 production (Figure 1). In particular, p24 production was similar up to day 7 post infection for both CCR5- and CXR4-tropic strains. After 7 days post infection, p24 production of the CCR5-tropic 81A virus underwent a sharp increase up to day 10 and then tended to remain stable up to day 14. Conversely, p24 production of the CXCR4-tropic NL4.3 sharply decreased, becoming almost undetectable starting from day 10 (Figure 1). As a control, pre-treatment with AMD3100 completely abrogated the replication of the CXCR4- tropic NL4.3 in MDM.

### 3.2. Effect of CXCR4-Tropic NL4.3 and/or CCRR5-Tropic 81.A Dependent HIV Infection Upon DNA Fragmentation

The second step of this study was to investigate the impact of CCR5-tropic 81.A and CXCR4-tropic NL4.3 on inducing apoptosis in MDM by cytofluorometry. In MDM infected with the CXCR4-tropic NL4.3, the percentage of cells in apoptosis progressively increased from day 4 up to day 10 post infection (43.5%) (time point at which the p24 production of the CXCR4-tropic NL4.3 becomes undetectable) and tended to remain stable at around 40% 13 days post infection (Figure 2). As a control, pre-treatment with AMD3100 abrogated the capability of CXCR4-tropic NL4.3 to induce MDM apoptosis (Figure 2). Conversely, very low levels of apoptosis were observed in MDM infected with the CCR5-tropic 81A. Overall findings suggest that the decrease in the replication of the CXCR4-tropic NL4.3 in MDM may be linked to the capability of the viral strain to induce apoptosis of MDM. The capability of CXCR4-tropic strains to favor MDM apoptosis was confirmed in the presence of another laboratory adapted CXCR4-tropic IIIB and in the presence of different CXCR4-tropic clinical isolates. Interestingly, the CCR5/CXCR4-tropic clinical isolate #6 conserved the capability to induce MDM apoptosis despite dual tropism (Figure 3).

### 3.3. Impact of CXCR4-Tropic (CXCR4) and CCR5-Tropic (CCR5) Strains in p38 Activation in MDM

To further corroborate the capability of CXCR4-tropic strains to favor MDM apoptosis, the detection of the phosphorylated form of the mitogen-activated protein p38/MAPK (phospho- p38) was evaluated. After exposing NL4.3 for 30 min (Figure 4 Lane 9), a 2-fold increase in the detection of the phospho- p38 MAPK was observed compared to MDM exposed to p81A. Thus, NL4-3 induced the activation of the p38/MAPK since early phases of MDM infection (while no effect was observed for 81A) [62]. Similarly, an increase in the detection of the phosphorylated form of p38 was also observed after exposing the CXCR4-tropic IIIB to MDM for 30 min (Figure 5A). Conversely, the detection of the phosphorylated form of p38 was reduced when AMD3100 was used (Figure 5B).

### 3.4. Modulation of Expression of Genes Correlated with Apoptotic Pathways, in a Time-Dependent Manner, by CXCR4-Tropic Strains

As a final step of this study, we analyzed variation in the transcriptome profile observed in MDM infected by CCR5-tropic or CXCR4-tropic strains. We found that NL4-3, but not 81A, up-regulated many genes in human MDM, including TNF2, Fas (TNFRSF6)-associated via death domain [63], caspase-7, Cytocrome C, KGF [64] and GSPT1/eRF3 [65] but down-regulates survival and cancer genes. These findings indicate that CXCR4-mediated the entry in MDM can up regulate apoptosis-related genes and simultaneously down modulate survival-related genes as Defender against cell death 1(DAD-1) and Cullin 2 (hCUL2), involved in MDM survival after HIV-1 infection. Interestingly, we also observed enhancement in gene activation of pro-inflammatory Matrix metalloproteinase 9 (MMP9 gelatinase B, 92kD gelatinase, 92kD type IV collagenase) by the CXCR4-tropic NL4.3 but not by CCR5-tropic p81.A (Figure 6 and Appendix A).

## 4. Discussion

This study shows that CCR5-tropic and CXCR4-tropic strains exhibit different kinetics of replication in MDM and highlights the capability of CXCR4-tropic strains to promote the apoptosis of this important HIV-1 reservoir. HIV-1 reservoirs represent so far the major obstacle for achieving HIV cure.

Our findings are in line with a previous study showing that differences between CCR5-tropic strains and CXCR4 strains in productive infection of MDM occurred during the early stages of HIV-1 life cycle and in particles at levels of reverse transcription and nuclear translocation of viral genomes [32]. Though our study did not consider the phase22s of HIV life cycle, we investigated a relationship between different levels in viral production and MDM homeostasis according to co-receptor usage.

We evaluated different kinetics of replication in MDM of CXCR4-tropic and CCR5-tropic molecular clones, respectively, NL4-3 and 81A, differing only in env variable domains. After a starting boost, the replication of CXCR4-tropic clones in MDM subsequently diminished reaching a status of abortive infection, while the replication of CCR5-tropic clones tended to increase, reaching a plateau after 10 days of infection.

It is important to stress that NL4-3 did not affect HIV-1 productive infection up to the seventh day in MDM, suggesting that the clearance of the CXCR4 strain may not be due to a failure in the entry or in other preintegrational phases (Figure 1), but may be the result of the killing of the host cells during the onset of infection.

These results underline the tendency towards an in vitro disappearance of the most aggressive CXCR4- tropic virus in the course of the HIV-1 infection and the survival of CCR5- tropic strain infected MDM reservoirs as key determinant of HIV-1 persistence in this cellular reservoir.

This evidence provided us the clue to analyze how coreceptor usage may differently modulate MDM homeostasis and particularly apoptosis.

In particular, the MAPK p38 plays a pivotal role in the transmission of signals from cell surface receptors to the nucleus. It is activated by diverse extracellular stimuli that regulate important cellular processes including response to stress factors in many cell types [66]. Our results show a transient but marked induction of the phosphorylated form of the p38 MAPK at 30′ after exposure to CXCR4-tropic, but not CCR5-tropic HIV strains in MDM. The role of activation of MAPKs p38 in programmed death of MDM and T-cells due to CXCR4-tropic strain infection still remains controversial: whereas a role in HIV pathogenicity is already demonstrated [50,67], some studies report no association in Caspase-dependent apoptosis [68] moreover, the p38 activation pathway, in cell reservoirs such as MDM, was attributed to β chemokine secretion rather than apoptosis [49,69]. In this last case, this disagreement with our results in HIV mediated signalling may be attributed to a different experimental approach as we used the whole pure viruses and not the recombinant gp120 and did not consider any serum starvation for exposure of MDM to the different strains, in order to avoid them excessive stress and permit the primary cells to reproduce under more natural physiological conditions.

The role of p38 has been elucidated in the setting of infection of T cells by CXCR4-tropic strains [70,71]. In particular, the role of replication of HIV-1 in human T lymphocytes requires the activation of host cellular proteins [72]. Previous studies have identified p38 mitogen-activated protein kinase (MAPK) as a kinase necessary for HIV-1 replication in T cells [73,74,75,76]. Among them, Cohen et al. 1997 have shown that HIV-1 CXCR4 strain infection of both primary human T lymphocytes and T cell lines immediately stimulates the cellular p38 MAPK pathway, which remains activated throughout the experimental conditions. Inclusion of an antisense oligonucleotides to p38 MAPK expressly inhibited viral replication [70,77,78,79]. Blockade of p38 MAPK by addition of CNI-1493 also inhibited HIV-1 viral replication of primary T lymphocytes in a dose- and time-dependent manner. Stimulation of p38 MAPK activation did not occur with the addition of heat-inactivated virus, suggesting that viral internalization, and not just membrane binding, is necessary for p38 MAPK activation [80,81]. The results of this work show that activation of the p38 MAPK cascade is critical and essential for HIV-1 replication in T cells [81,82].

In consideration of key determinants of HIV persistence in MDM reservoirs, post translation changes of cell and nuclear targets is one of the upstream events due to viral exposure, culminating in absence of cytolitic effects. Macrophages provide an ideal environment for the formation of viral reservoirs since they live long and are widely distributed throughout the body [83].

As microarray analysis showed, 10 genes related to the apoptosis pathway were up-regulated in NL4-3 infected MDM compared to 81A infected ones and genes related to the apoptosis pathway, such as Defender against cell death 1(DAD-1) [84] and Cullin 2 (hCUL2), were up-regulated in 81A infected compared to NL4-3 infected MDM.

Our studies demonstrate the up-regulation of genes included the polypeptide chain-releasing factor GSPT1/eRF3 protein, which in the processed form has been shown to promote caspase activation, IAP (inhibitors of apoptosis) ubiquitination and apoptosis [53], Caspase 7 (CASP 7), an apoptosis-related [69] and Cytochrome C whose release has been shown in HIV dependent apoptosis [50]. The activation of such genes related to apoptosis in NL4-3 infected MDM can then have downstream effects being responsible for the progressive decrease of p24 production of the CXCR4-using NL4.3.

Our results would not seem in line with in vivo evidence of the emergence of the more aggressive sincytium-inducing (SI) CXCR4- tropic strains in the terminal phases of HIV-1 disease associated with rapid decline of CD4+ and CD8+ T cells [28,85] but this phenomenon almost represents an effect of the breakdown of the immune system and the onset of AIDS [86]. We speculate that CXCR4- tropic strains play a minor role in disease progression because dying CXCR4 virus infected reservoirs, cannot provide virus nor continue to directly contribute to the depletion of immune cell system. Indeed, CCR5-using strains are associated with a lower percentage of cell death, suggesting the capability of these strains to promote cell survival as supported also by transcriptome analysis.

On the other hand, there are many reasons to consider a role of CCR5 viruses and their host cells as target for therapeutic strategies: (i) the protective role of the 32-nucleotides (Δ32) deletion in CCR5 gene in homozygous condition against HIV-1 infection and the more benign pattern of disease progression associated with the deletion in one of the two alleles [87,88] (ii) a logarithmic correlation between CCR5 expression and viremia in patients with disease progression [89,90,91] (iii) the importance of CCR5-tropic isolates for dissemination outside peripheral blood in compartments considered as “sanctuaries” like the Central Nervous System where macrophages represent more than 90% of the HIV-1 infected cells [92,93,94,95] the capacity of R5 isolates harbored in macrophagic reservoirs to provoke the immune anergy through host-related factors (bystander effect) and the emergence of more virulent SI variants and the subsequent AIDS progression (iv) increase of viremia in later stages of HIV disease caused by macrophages during opportunistic infections [96,97] (v) increase of both CCR5 expression on CD4+ T cells and the frequency of memory CD4 T-cells (the target cells of CCR5 virus variants) over the course of infection [97,98].

## 5. Conclusions

To summarize, CCR5 strains induce chronic and productive infection in MDM whereas CXCR4-tropic strains induce abortive infection. Moreover, the abrogation of HIV-1-dependent killing due to the specific CXCR4 inhibitor AMD3100 indicates the obligatory role of CXCR4. Phosphorylation of p38 (MAP Kinase family), reported to be activated after exposure to many forms of cellular stress, is enhanced by the CXCR4-tropic strain NL4-3 and IIIB and not by CCR5-tropic strain 81A and BaL; also, this induction is modulated by CXCR4. CXCR4-tropic strains activate inflammatory genes in MDM whereas CCR5-tropic HIV-1 strains do not induce a death program in MDM.

Taken together, our results correlate with in vitro and in vivo evidence about an uncoupling between viral replication and cytopathocity [88] and confirm what we observed previously [9,99] in later phases of infection: MDM homeostasis is up-regulated where infection is sustained by CCR5-tropic strains. All the CXCR4-tropic strains we used, in contrast, induce MDM apoptosis and lead to consequent clearance of HIV replication. Further studies are necessary to investigate if gp120 interaction with CXCR4 in MDM can also induce the activation of pro-inflammatory pathways.

Our findings provide important implications for HIV-1 pathogenesis and design of pharmacological targets aimed at achieving HIV-1 cure.

## Figures and Tables

**Figure 1 medicina-55-00297-f001:**
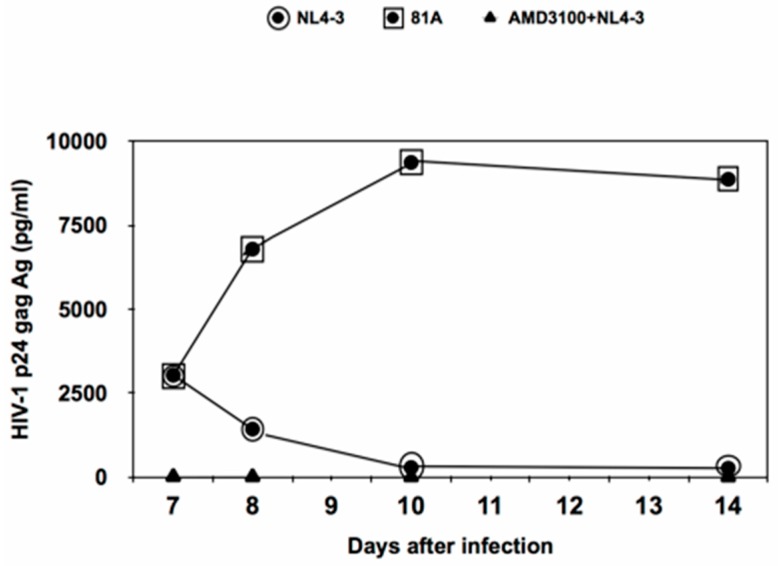
Kinetics of the HIV-1 replication profile in human primary monocyte-derived macrophages (MDM) infected by CXCR4-tropic NL4.3 and or CCR5-tropic 81.A virus. The figure reports p24 production starting from day 7 post-infection. MDM from a healthy donor were infected with a standard dose of p24 gag (3000 pg/mL) of NL4.3 or 81.A. p24 production was measured daily in culture supernatants by a commercially available ELISA (Abbott labs, Pomezia, Italy). Pre-treatment with AMD3100 was performed 1 h before incubation with NL4.3. All tests were performed in triplicate.

**Figure 2 medicina-55-00297-f002:**
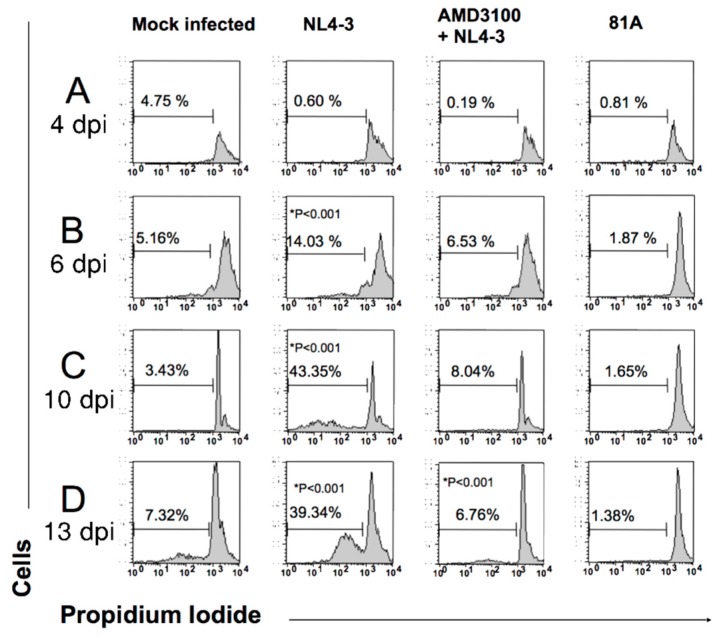
Effects of CXCR4 and CCR5 usage on DNA fragmentation (measure of apoptosis) in MDM. MDM from a healthy donor were infected with 81A (CCR5-tropic strain) or NL4-3 (CXCR4-tropic strain). Percentage of cells undergoing DNA fragmentation was checked: (**A**) 4 days after infection (**B**) 6 days after infection; (**C**) 10 days after infection; (**D**) 13 days after infection. Pretreatment with AMD3100 was performed 1 h before incubation with NL4-3. For HIV infection in MDM, a standard dose of p24 gag (3000 pg/mL) was used. The PI fluorescence was measured by Flow Cytometry in FL2-H (FACScan, Becton Dickinson System, San josè, CA) and registered on a logarithmic scale. The figure is representative of three independent experiments. Differences in NL4-3-infected macrophages are statistically significant (*P* < 0.001, Chi Square test) compared to mock-, 81A-infected and AMD3100- treated macrophages.

**Figure 3 medicina-55-00297-f003:**
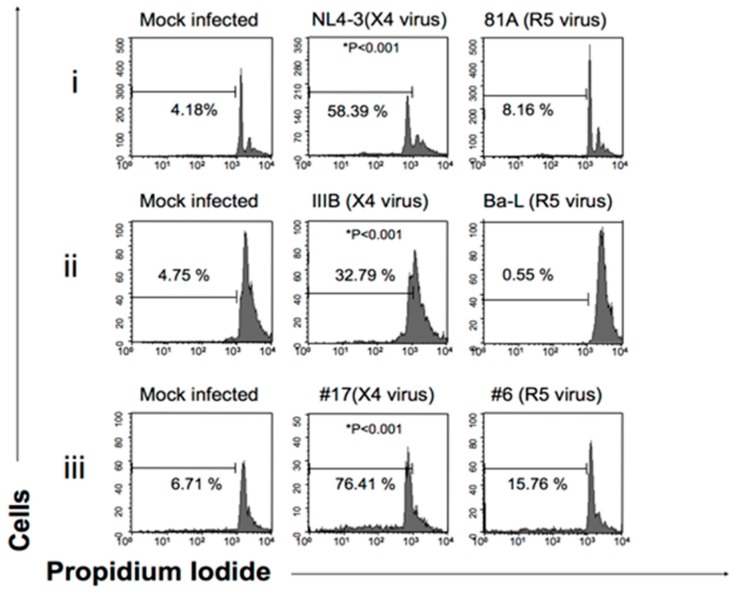
Measure of DNA fragmentation in Human Primary Macrophages infected by X4-tropic virus. The PI fluorescence was measured by Flow Cytometry in FL2-H and registered on a logarithmic scale. All the tests were performed in duplicate. (**i**) NL4-3 and 81A are, respectively, a CXCR4- (X4) and CCR5 (R5)-tropic HIV strains. DNA fragmentation was analyzed to the 10th day of infection (**ii**) IIIB and Ba-L are laboratory-adapted HIV strains using, respectively, X4 and R5 strains. DNA fragmentation was analyzed to the 7th day of infection. (**iii**) #17 and #6 are, respectively, a CXCR4- and CCR5-tropic HIV-1 clinical isolates and DNA fragmentation was analyzed to the 12th day of infection. For the infection, a standard dose of p24 gag (3000 pg/mL) correspondent to 400 TCID50/mL dose of Ba-L was used. * Differences in X4 strain infected macrophages are statistically significative (*P* < 0.001, Chi Squared test) compared to mock infected and R5 strain infected macrophages (see material and methods).

**Figure 4 medicina-55-00297-f004:**
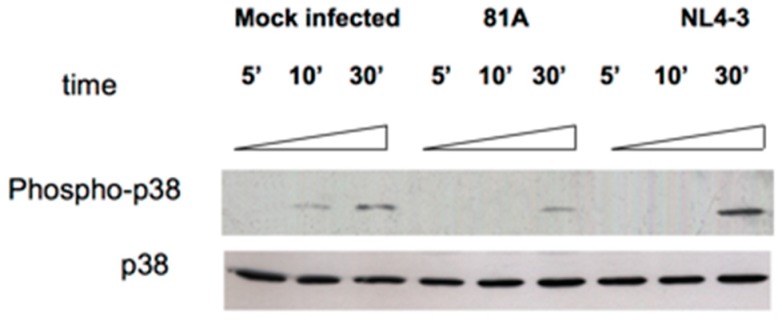
Detection of p38 and phospho p38 by Western Blotting in cell lysates. MDM were infected with 7500 pg/mL of NL4.3 or 81.A. Blots are representative of three experiments using MDM from different donors after exposing NL4.3 and 81.A in MDM for 5, 10 and 30 min.

**Figure 5 medicina-55-00297-f005:**
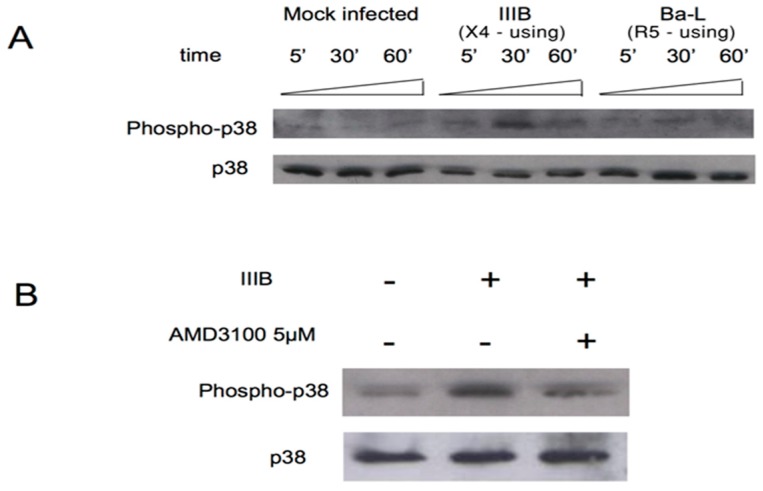
Detection of p38 and phospho p38 by Western Blotting. MDM were infected with 7500 pg/mL of the CXCR4-tropic IIIB and the CCR5-tropic 81.A. (**A**) Cell lysates were subjected to immunoblot analysis with antibodies specific for the total or phosphorylated forms of p38 MAPK (phospho-p38 MAPK [T180/Y182] antibody). (**B**) MDM from HIV-1 negative donor were pretreated for 30 min with (+) or without (−) AMD3100 (5µg/mL) before incubation with IIIB. Blots are representative of three experiments using MDM from different donors.

**Figure 6 medicina-55-00297-f006:**
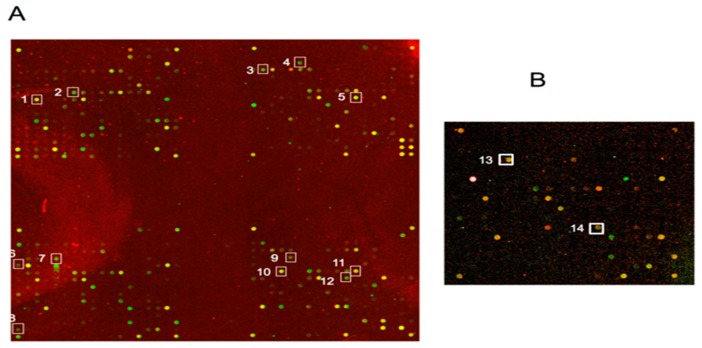
Photographs of arrays representing transcriptional changes in macrophages infected by 81A and NL4-3 strains. In array (**A**) the Cy3 spots indicated by numbered squares represent the apoptosis-related genes activated by NL4-3 whereas in subarray (**B**) the Cy5 spots indicated by numbered squares represent the survival-related genes activated by 81A. The square numbers relative to the genes are also indicated in Appendix A.

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
