# Peer review of "Different Patterns of HIV-1 Replication in MACROPHAGES is Led by Co-Receptor Usage"

_medicina, 2019, doi:10.3390/medicina55060297_

Round 1

Reviewer 1 Report

Anna and colleagues in the report describes the different fates of MDMs infected with two different tropic strains of HIV (CXCR4 and CCR5). Authors suggest that CCR5 using strains such as 81A efficiently replicates in MDMs as compared to CXCR4 using strain. The results of the study are important to understand the development and cure for HIV-1 latent reservoir such monocytes/macrophages. The introduction is well written. The method section provides sufficient details to understand the study design and information to reproduce the experiments. The study is well within the scope of the journal.

Major concern of the study are

1.       Authors reported, “CXCR4-using NL4.3 sharply decreased becoming almost undetectable starting from day 10”. However, in figure 6, the array data suggest changes in transcriptional profile of apoptotic genes at early time point. Please explain this contradictory observation.  

2.       Figure 3: middle panel: Does R5 infection help MDMs to survive as percentage PI labelled cells are more in mock-infected control as compared to R5 infected MDMs?

3.       Figure 4: phospho-p38 levels in mock infected were observed even after at 10 minutes. Authors should also consider some other marker of apoptosis such as protein levels of caspase7 (Figure 6, transcriptomic data).

4.       Since all of the experiments have been reproduced at least three times, therefore, I suggest to include the densitometry of the western blots.

5.       Figure 6: Some of the upregulated/downregulated transcripts could have validated by RT-qPCR. It will further strengthen the data.

Author Response

Anna and colleagues in the report describes the different fates of MDMs infected with two different tropic strains of HIV (CXCR4 and CCR5). Authors suggest that CCR5 using strains such as 81A efficiently replicates in MDMs as compared to CXCR4 using strain. The results of the study are important to understand the development and cure for HIV-1 latent reservoir such monocytes/macrophages. The introduction is well written. The method section provides sufficient details to understand the study design and information to reproduce the experiments. The study is well within the scope of the journal.

Major concern of the study are:

  Authors reported, “CXCR4-using NL4.3 sharply decreased becoming almost undetectable starting from day 10”. However, in figure 6, the array data suggest changes in transcriptional profile of apoptotic genes at early time point. Please explain this contradictory observation.  

In our work, the different replicative kinetics of CXCR4- and CCR5-dependent HIV strains in MDM were studied. We find that p24 production of the CXCR4-using NL4.3 sharply decreased becoming almost undetectable starting from day 10 (Figure 1) In the Figure 6, the transcriptional changes are studied at early time point because they take place in the earliest stages of HIV infection. This is the reason why the arrays were done in 81A and NL4-3 infected MDMs (6 to 24 hours of infection). The activation of such genes related to apoptosis in NL4-3 infected MDM can then have downstream effects being responsible for the progressive decrease of p24 production of the CXCR4-using NL4.3.

We have added this concept in the discussion section of the revised version of the manuscript (page 11, line 42).

2.       Figure 3: middle panel: Does R5 infection help MDMs to survive as percentage PI labelled cells are more in mock-infected control as compared to R5 infected MDMs?

The figure 3 represents the measure of DNA fragmentation in Human Primary Macrophages infected by X4-using and R5-using virus. In this case, our experiments show that differences in the DNA fragmentation of X4 strain infected macrophages are statistically significant compared to mock and to R5 strain infected macrophages. We agree with the reviewer that CCR5-using strains are associated with a lower % of cell death, suggesting the capability of these strains to promote cell survival as supported also by trascriptome analysis.

We have added this concept in the discussion section of the revised version of the manuscript (page 11, line 50).

3.       Figure 4: phospho-p38 levels in mock infected were observed even after at 10 minutes. Authors should also consider some other marker of apoptosis such as protein levels of caspase7 (Figure 6, transcriptomic data).

We agree with the reviewer on the importance to test further markers of cell death. We detected cell death was PI staining and by measuring phospho-p38 (in 3 independent experiments), both are widely used in in vitro experiments to assess the rate of cell death. We are currently planning further studies to better elucidate the pathways associated with the death of CXCR4-infected macrophages. This will be the topic of our next studies.

4.       Since all of the experiments have been reproduced at least three times, therefore, I suggest to include the densitometry of the western blots.

We evaluated the expression of phospho-p38 in 3 independent experiments and obtained so far the same results. For this reason, we decided not to perform densiometry, but directly showed the raw data. However, it should be noted that results on cell death are further supported by PI and transcriptome analysis.

We appreciate the reviewer’s comment and apology for our mistake in reporting the use of Chi-squared test to verify statistical significance in figure’s legend. We have removed this sentence in the revised version of the manuscript.

5.       Figure 6: Some of the upregulated/downregulated transcripts could have validated by RT-qPCR. It will further strengthen the data.

The arrays represented in the Figure 6 have been performed to further corroborate results obtained in previous experiments (PI staining and phoshpo-p38 evaluation). We agree with the reviewer’s comment. Indeed, we have planned further experiments to elucidate in more details gene expression profiles in CXCR4-using strains infected macrophages. This will be the topic of our next studies.

Reviewer 2 Report

This manuscript describes an important role for co-receptor CCR5 or CXCR4 usage for infection and pathogenesis in MDM.  The retention of CCR5-tropic virus in MDM in the context of pathogenicity of CXCR4 strains has important implications in establishment of latent reservoirs and ultimate disease.  However, the paper is not clearly written and makes it difficult for the reader to follow the exactly what was done and understand its implications.  Extensive rewrite will be needed to clear up the language.

Specific Comments

AMD3100 in Fig 1 shows that there is no CXCR4 virus produced.  Is  the dose of AMD3100 toxic to cells, or specific for inhibition of CXCR4 strains?  Controls with CCR5 or cell toxicity are missing and it would be useful to discuss the specificity of the drug.

The figure 4 shows constant levels of p38 in the cells, but increased phosphorylation of p38 with time in both infected and R5 and X4 infected cells.  It is difficult to visually interpret the differences in levels of p38 under these three conditions, or those used in Fig 5.  How was the level quantitated, and what were the bases for significance p-value calculations? 

Some additional discussion of the role of p-38 in apoptosis, perhaps with additional references would also be helpful in understanding its role in virus mediated pathogenicity.

Fig 6 is difficult to understand, and the Supplementary table is helpful.  However, how it would be helpful to understand how the dots were quantitated, and the extend of inhibition or enhancement of these proteins following infection.

I would suggest the use of CCR-5 or CXCR4-tropic rather than -using to be more in keeping with the literature.

Author Response

This manuscript describes an important role for co-receptor CCR5 or CXCR4 usage for infection and pathogenesis in MDM.  The retention of CCR5-tropic virus in MDM in the context of pathogenicity of CXCR4 strains has important implications in establishment of latent reservoirs and ultimate disease. However, the paper is not clearly written and makes it difficult for the reader to follow the exactly what was done and understand its implications. Extensive rewrite will be needed to clear up the language.

Specific Comments:

AMD3100 in Fig 1 shows that there is no CXCR4 virus produced. Is the dose of AMD3100 toxic to cells, or specific for inhibition of CXCR4 strains? Controls with CCR5 or cell toxicity are missing and it would be useful to discuss the specificity of the drug.

AMD3100 has a powerful inhibitory activity against X4 viruses and is a potent and selective antagonist of CXCR4 receptor; it has been shown to block the route of entry of HIV into host T-cells by several studies since several years (-Donzella GA1, Schols D, Lin SW, Esté JA, Nagashima KA, Maddon PJ, Allaway GP, Sakmar TP, Henson G, De Clercq E, Moore JP. AMD3100, a small molecule inhibitor of HIV-1 entry via the CXCR4 co-receptor. Nat Med. 1998 Jan;4(1):72-7;

- Schols D, Struyf S, Van Damme J, Esté JA, Henson G, De Clercq E.Inhibition of T-tropic HIV strains by selective antagonization of the chemokine receptor CXCR4. J Exp Med. 1997 Oct 20;186(8):1383-8

- De Clercq E, Yamamoto N, Pauwels R, Balzarini J, Witvrouw M, De Vreese K, Debyser Z, Rosenwirth B, Peichl P, Datema R. Highly potent and selective inhibition of human immunodeficiency virus by the bicyclam derivative JM3100. Antimicrob Agents Chemother. 1994 Apr; 38(4):668-74;

- Kim HY, Hwang JY, Oh YS, Kim SW, Lee HJ, Yun HJ, Kim S, Yang YJ, Jo DY. Differential effects of CXCR4 antagonists on the survival and proliferation of myeloid leukemia cells in vitro. Korean J Hematol. 2011 Dec;46(4):244-52

- Knight JC, Hallett AJ, Brancale A, Paisey SJ, Clarkson RW, Edwards PG. Evaluation of a fluorescent derivative of AMD3100 and its interaction with the CXCR4 chemokine receptor. Chembiochem. 2011 Nov 25;12(17):2692-8.

Furthermore, AMD 3100 is not toxic at concentration of >100µM in MDM (J Leukoc Biol. 1997 Jul;62(1):138-43) and has been used in our laboratory for many years in primary culture cells of lymphocytes and macrophages and we have a standardized technique for this procedure as reported in : -(Surdo M, Balestra E, Saccomandi P, Di Santo F, Montano M, Di Carlo D, Sarmati L, Aquaro S, Andreoni M, Svicher V, Perno CF, Ceccherini-Silberstein F. Inhibition of dual/mixed tropic HIV-1 isolates by CCR5-inhibitors in primary lymphocytes and macrophages. PLoS One. 2013 Jul 9;8(7):e68076).

- Aquaro S, Perno CF, Balestra E, Balzarini J, Cenci A, Francesconi M, Panti S, 

Serra F, Villani N, Caliò R. Inhibition of replication of HIV in primary monocyte/macrophages by different antiviral drugs and comparative efficacy in

lymphocytes. J Leukoc Biol. 1997 Jul;62(1):138-43.

The figure 4 shows constant levels of p38 in the cells, but increased phosphorylation of p38 with time in both infected and R5 and X4 infected cells.  It is difficult to visually interpret the differences in levels of p38 under these three conditions, or those used in Fig 5. How was the level quantitated, and what were the bases for significance p-value calculations? 

The objective of the analysis was to evaluate differences in the phosphorylated form of p38 since this is the form associated with the activation of apoptosis. Indeed, we expected no differences in p38 but in phospho-p38. The Western Blotting was performed in 3 independent experiments (cells obtained from 3 different donors) and we decided to directly show raw data. It should also be noted that the impact of CCXCR4-using strains on death of macrophages was also supported by PI staining and transcriptome analysis.

We thank the reviewer and we apology for the mistake to have reported the use of Chi-squared test in figures’ legends. We have removed such sentence accordingly.

Some additional discussion of the role of p-38 in apoptosis, perhaps with additional references would also be helpful in understanding its role in virus mediated pathogenicity

    We thank the reviewer for giving us the opportunity to better explain the role of p38 that has been elucidated in the setting of infection of T cells by CXCR4-using strains. In particular, the role of replication of HIV-1 in human T lymphocytes requires the activation of host cellular proteins. Previous studies have been identified p38 mitogen-activated protein kinase (MAPK) as a kinase necessary for HIV-1 replication in T cells (Cohen et al.1997).

Among them, Cohen et al. 1997 have shown that HIV-1 CXCR4 strains infection of both primary human T lymphocytes and T cell lines rapidly activates the cellular p38 MAPK pathway, which remains activated throughout the entire experimental conditions. Addition of an antisense oligonucleotides to p38 MAPK specifically inhibited viral replication. Blockade of p38 MAPK activation by addition of CNI-1493 also inhibited HIV-1 viral replication of primary T lymphocytes in a dose- and time-dependent manner. Stimulation of p38 MAPK activation did not occur with the addition of heat-inactivated virus, suggesting that viral internalization, and not just membrane binding, is necessary for p38 MAPK activation. The results of this work indicate that activation of the p38 MAPK cascade is critical for HIV-1 replication in T cells.

This has been included in the discussion of the revised version of the manuscript.

Other studies addressing the role of p38 in HIV infection are as follows:

- Bai L, Zhu X, Ma T, Wang J, Wang F, Zhang S. The p38 MAPK NF-κB pathway, not the ERK pathway, is involved in exogenous HIV-1 Tat-induced apoptotic cell death in retinal pigment epithelial cells. Int J Biochem Cell Biol. 2013 Aug;45(8):1794-801.

- Chung TW, Choi H, Lee JM, Ha SH, Kwak CH, Abekura F, Park JY, Chang YC, Ha KT, Cho SH, Chang HW, Lee YC, Kim CH. Oldenlandia diffusa suppresses metastatic potential through inhibiting matrix metalloproteinase-9 and intercellular adhesion molecule-1 expression via p38 and ERK1/2 MAPK pathways and induces apoptosis in human breast cancer MCF-7 cells. J Ethnopharmacol. 2017 Jan 4;195:309-317.

- Kralova J, Dvorak M, Koc M, Kral V. p38 MAPK plays an essential role in apoptosis induced by photoactivation of a novel ethylene glycol porphyrin derivative. Oncogene. 2008 May 8;27(21):3010-20.

-  Bai L, Zhu X, Ma T, Wang J, Wang F, Zhang S. The p38 MAPK NF-κB pathway, not the ERK pathway, is involved in exogenous HIV-1 Tat-induced apoptotic cell death in retinal pigment epithelial cells. Int J Biochem Cell Biol. 2013 Aug;45(8):1794-801.

Fig 6 is difficult to understand, and the Supplementary table is helpful. However, how it would be helpful to understand how the dots were quantitated, and the extend of inhibition or enhancement of these proteins following infection.

Fluorescent-array images were collected for both Cy3 and Cy5 by using a ScanArray Express, Microarray Analysis System Version 2.0 (Perkin-Elmer) and image intensity data were extracted and analysed by using QuantArray Pachard Biochips Software. In particular, QuantArray Software provides automated analysis of color microarray images (automatic scanning and quantitation to measure fluorescence signal at each spot on the array) before exporting data to bioinformatics software packages. Triplicate array positions are used for each gene to avoid signal noise. In Human Cancer Chip used for microarray analysis, all spots were known. In order to evaluate the inhibition or enhancement of genes expression in terms of mRNA production, the comparison of Cy3 and Cy5 signals intensity were applied. This has been included in the material and methods of the revised version of the manuscript.

I would suggest the use of CCR-5 or CXCR4-tropic rather than -using to be more in keeping with the literature.

According with the reviewer’s comment, we have changed the nomenclature of the strains.

Round 2

Reviewer 1 Report

I have gone through the responses made by authors. I agree with them, some of the key experiments can be the part of their future studies. Rest, all of the responses are satisfactory. I congratulate authors for the good work. This is an important contribution in the area of HIV-1 biology. 

Recommendations: Accept

Author Response

Thank you for your comments and suggestions that allowed us to greatly improve the quality of the manuscript. We agree with all your comments, and as you proposed, we have made changes of English language in the manuscript (all changes that were made, appear in yellow in the final version of the paper).

Reviewer 2 Report

I would like to thank the authors for their revisions and clarification which addressed the previous concerns.  The paper now reads much better, and is acceptable for publication.  I still suggest some editing to make the manuscript clearer.

Author Response

Thank you for your very careful review of our paper, and for the comments, corrections and suggestions that ensued. We believe the paper has been significantly improved.

In the final version of the manuscript, we detail the editing changes that you have suggested. (All changes that were made, appear in yellow in the final version of the paper).
